# Printed Graphene Electrode for ITO/MoS$_2$/Graphene Photodiode Application

Amal M. Al-Amri [1,*], Tien Khee Ng [2], Nour El I Boukortt [3] and Boon S. Ooi [2]

1    Physics Department, Collage of Science & Arts, King Abdulaziz University, Rabigh 25724, Saudi Arabia
2    Photonics Laboratory, Computer, Electrical and Mathematical Sciences and Engineering (CEMSE) Division, King Abdullah University of Science and Technology (KAUST), Thuwal 23955, Saudi Arabia
3    Electronics and Communication Engineering Department, Kuwait College of Science and Technology, Doha 13113, Kuwait
*    Correspondence: amsalamri@kau.edu.sa

**Abstract:** Lightweight and flexible electronics have recently emerged at the forefront of optoelectronic applications. In this regard, graphene electrodes enable opportunities for new photodiode devices. In this paper, we formulated and tested graphene ink using the standard inkjet printing technique. It was shown that the maximum conductivity of ink was achieved for 14 print passes of the graphene layer. Moreover, we deposited Molybdenum Disulfide (MoS$_2$) ink in the same pattern and used it as an active layer. We put MoS$_2$ ink on an Indium-Tin-Oxide (ITO) glass substrate and then deposited graphene ink as a top electrode to fabricate an ITO/MoS$_2$/graphene device. The fabricated device showed good rectification behavior and high ON/OFF switching behavior with a max photocurrent of 15 μA at +2 V. The technique thus paves the way for low-cost, low-temperature processing of electronics and one-step fabrication.

**Keywords:** graphene; diode; photocurrent; inkjet printing; spectroscopy; UV-Vis; percolation theory

## 1. Introduction

Manufacturing techniques in the consumer electronics industry have made a lot of progress in the last decade, allowing for the creation of smaller, faster, and more efficient devices for common use. However, these devices' flexibility, environmental impact, and processing cost are all impacted by using standard solid-state technology. Hence, devices in the consumer electronics industry have undergone a remarkable evolution in recent years, reducing in size while also becoming more flexible and suitable for wearable applications. Lightweight and flexible electronics have recently emerged, finding use in emerging markets such as internet of things (IoTs) and wearable electronics. Due to many factors [1–3], printing flexible electronics has emerged as a viable replacement for the conventional production of inorganic materials. At the same time, printing techniques provide low-cost and simple methods for device fabrication while showing compatibility with most substrates, including soft and flexible ones [1].

There is tremendous growth potential for the organic and printed electronics that are currently available. Printing conductive inks on flexible substrates paves the way for this shift because it is cheap, easy, and scalable to mass produce devices with high flexibility and stretchability [4]. Inkjet printers can be used to create a multilayer printed circuit board (PCBs) and electrodes. Moreover, inkjet printing is an additive, noncontact printing technique that provides excellent throughput, low waste, and processability [2].

Two-dimensional materials have emerged as one of the most suitable alternatives in recent years due to their superior characteristics, which include electrical, optical, and even mechanical properties. Notably, their recent advances in realizing practical uses have been inspired beyond single-device realization, launching advanced research for low-cost, large-area, and high-yield rapid manufacturing. In this regard, ubiquitous inkjet

printing has been used as a viable solution to meet large area requirements, low cost, ease of processability, and high compatibility. Furthermore, inkjet printing can provide 2D nanoelectronics with a high degree of design freedom while maintaining high electrical performance. Graphene and other 2D materials have recently attracted the scientific community's attention for various electronic applications. These materials, including boron nitride (BN), manganese dioxide ($MnO_2$), and molybdenum disulfide ($MoS_2$), have many valuable properties. Indeed, such materials are likely to be used as active materials in a wide range of printed device applications.

Furthermore, it is expected that several applications will demand the possibility of printing combinations of 2D materials, for example, a conducting material such as graphene as an electrode and a semiconducting material such as $MoS_2$ as the active element. Graphene, composed of a single layer of sp2-hybridized carbon arranged in a honeycomb lattice, has attracted much scientific interest because of its unusual chemical and physical characteristics. Graphene's remarkable mechanical strength and high in-plane thermal conductivity are due to the three s-bonds formed by the sp2 bonding between its carbon atoms. Moreover, graphene has enhanced the electrical properties of $MoS_2$ composites [5]. There has been extensive use of bulk $MoS_2$ or graphene in Schottky devices in the realm of the metal–semiconductor interface. Due to the high tunability of the Fermi level and the stable excitonic states of monolayer $MoS_2$ at ambient conditions, the addition of graphene makes the $MoS_2$/graphene composite into a nearly perfect platform to reveal a 2D semiconductor and 2D semimetal junction. This makes them practical for optoelectronic and photonic device applications [6,7].

An essential step toward increasing the application of graphene-based technologies is the industrial production of graphene thin films on a large scale. The unique electronic, optical, and mechanical properties of graphene make inkjet printing of graphene [2] a vital research path in these areas. This is because it adds the attractive features of inkjet printing (low cost, direct writing, additive patterning, and scalability to large-area production) [8,9]. Many different nanomaterials have been successfully deposited using this method [10–12]. Printing on graphene improves device performance and conductivity, but it requires high-temperature chemical treatment. These techniques not only harm the print surfaces but also minimize the material's flexibility (plastic film, paper). The results of the experiments showed that graphene could be used to make wearable electronics that are both flexible and inexpensive. The inkjet printing method of deposition is cheap and can be used to create either transparent or opaque conductive films. Printed graphene, also known as reduced graphene oxide (rGO), has many potential uses. These include sensors, antennas, transparent conductors, thin film transistors, supercapacitors, optoelectronic devices, photodiodes, photodetectors, solar cells, and many more [13–17].

Graphene electrodes enable opportunities for new optoelectronic devices. Large-area, residue-free graphene film can be used as a transparent conducting electrode in flexible devices such as organic photovoltaic cells (OPVs), organic light-emitting diodes (OLEDs), and photodetectors (PDs). Ideal optoelectronic electrodes have high optical transparency, low sheet resistance, and the necessary work function. Fabrication costs must be considered for a profitable commercial release. Moreover, ITO has several drawbacks that prevent it from being used in flexible devices. Flexible optoelectronics need transparent conducting electrodes. Graphene's electrical and mechanical properties make it a promising candidate in comparison to ITO electrodes. Many researchers have attempted to use graphene electrodes in OPVs and OLEDs [18,19]. In [18], monolayer graphene is used as a transparent electrode to fabricate a semitransparent and flexible organic solar cell (OSC). The graphene electrode showed good transmittance at 700 nm wavelength, and the device achieved 14.2% power conversion efficiency (PCE). The authors of [19] created a chemical vapor deposition (CVD)-grown, graphene-based, transparent, multilayered graphene electrode for fabricating efficient OSC. The tested OSC exceeded 92% optical transmittance with 16% PCE. Moreover, OLEDs rely heavily on transparent conducting electrodes (TCEs). Constructing a stable, low-cost, flexible TCE for next-generation OLED-based displays

is a major step forward in this field. Since graphene maintains its electrical properties even when bent at a radius on the millimeter scale, it is a promising candidate for use as a flexible TCE. Organic light-emitting diodes with TCEs made of graphene have been the subject of a lot of research. Chen et al. [20] realized an efficient nanorod array-based LED by utilizing a transparent graphene electrode. The authors of [21] also utilized CVD-synthesized, multilayered graphene as a diode. The pristine graphene worked as a p-type material, while the n-type material was tuned by nitrogen plasma treatment. Adetayo et al. [22] presented a detailed review of the effectiveness of graphene as a potential electrode for OLEDs. Printing graphene is also useful for making photodetectors with a Graphene/Quantum Dot (Graphene/QD) heterostructure [23]. The low cost, variety, and simple processing of printed graphene make it ideal for this application. Printable graphene may be shaped into complex geometries, making it ideal for the large-scale integration of electronic components.

However, the existing fabrication techniques require high processing temperatures and costly mechanisms. Furthermore, the available electrodes are costly and rigid, limiting their application in flexible electronics. This paper focuses on addressing these issues by incorporating an inkjet printing technique for graphene electrode fabrication. Inkjet printing can overcome challenges in traditional fabrication processes, producing new functionalities and improving existing processes. However, some challenges exist in producing high-quality ink using inkjet printing techniques. The nanoparticles in ink should be adequately dispersed with the required viscosity to ensure uniform printing of patterns. Moreover, the surface tension of the ink should be controlled to avoid nonuniform film deposition. In this work, we have focused on achieving low surface tension (31 mN/m) and the desired viscosity (7.5 cP) of the ink by exfoliating graphene nanoparticles. Hence, the graphene ink showed good precision and resolution. Then we optimized the conductivity of the graphene electrode. This helped us in tuning the work function of the electrode and its utilization as a p-type material for making a p–n junction photodiode. Then the inkjet printing technique was utilized to deposit the active material $MoS_2$ on an ITO electrode to fabricate an $ITO/MoS_2/graphene$ device. These techniques can overcome challenges in conventional device fabrication, such as water-based electron lithography, and improve optoelectronic device performance. The developed electrode based on graphene ink is low-cost and scalable for printed electronics and photonics applications. We have fabricated a novel p–n junction device and achieved good rectification with a high ON/OFF ratio.

## 2. Materials and Methods

### 2.1. Material Preparation

In this work, the ink formulation technique was chosen because it yields the most desirable results for the exfoliation of graphene and other 2D materials. Initially, graphite powder (purchased from Sigma Aldrich, St. Louis, MO, USA) in dimethylformamide (DMF) was sonicated for 24 h at room temperature, as shown in Figure 1. Once the solution was obtained, it was centrifuged to separate the thick flakes in the sediment from the supernatant. Then a stable graphene dispersion was obtained by adding ethylcellulose to the mixture. Then the DMF was removed from the solution (terpineol 2 mL + 0.2 mL ethelo polymer) through a rotary evaporator equipped with vacuum distillation. At 80 °C, DMF began to evaporate from the dispersion. Once the DMF was removed via boiling, the residual graphene/terpineol dispersion was collected. The concentration of graphene in the final terpineol dispersion was higher than in the first DMF dispersion. Then a sample of the ink solution was diluted and analyzed for its optical absorbance to determine the graphene content. Finally, the graphene ink was heated to 400 °C inside a nitrogen-filled oven for one hour to achieve the standard viscosity for inkjet printing.

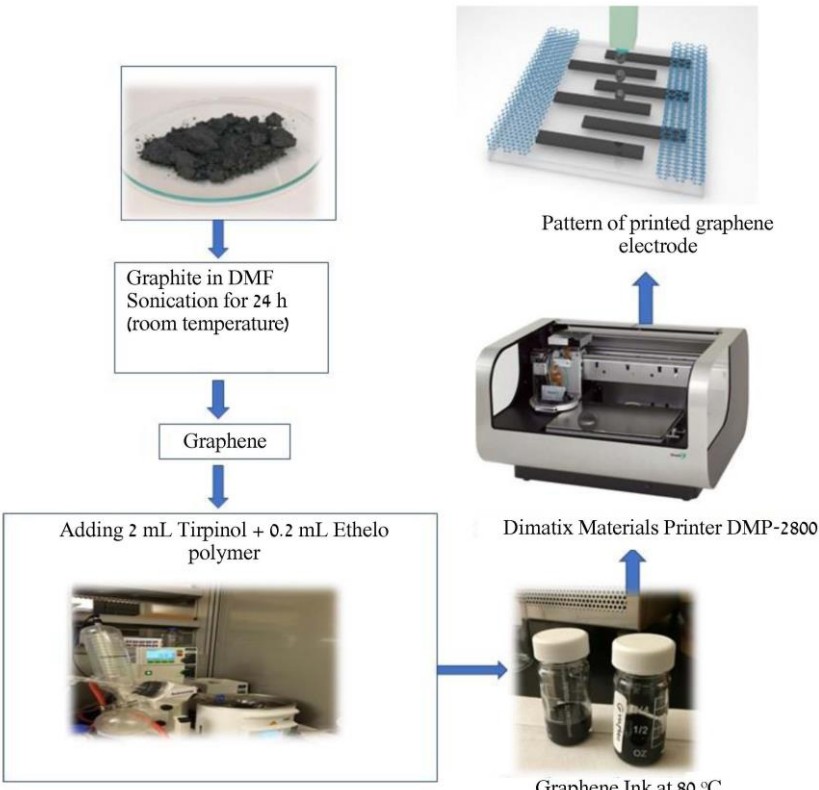

**Figure 1.** Schematic diagram of graphene ink formulation and equipment.

## 2.2. Device Printing

To thoroughly clean glass substrates, ultrasonication was performed in acetone, 2-propanol, and finally, deionized water. After being dried with a stream of nitrogen, the substrates were treated for 30 s with oxygen plasma. A piezo inkjet cartridge was used by employing a commercial piezoelectric Fujifilm Dimatix Material Printer (DMP-2800, Way College Station, TX, USA), to print the interdigitated electrodes on a glass substrate. We employed graphene inks, both developed in-house and black inks available in the market. The annealing procedure entails nothing more complicated than placing the components on a baking sheet and placing it in a preheated oven at 400 °C for 1 hr. As seen in the Advanced Design System Software (version 2022), the interdigitated contact electrodes were printed in a square array with varying spacing (150, 200, and 250 μm), as shown in Figure 2. The length (L) and width (W) of the printed electrode was 3.72 mm and 0.57 mm, respectively.

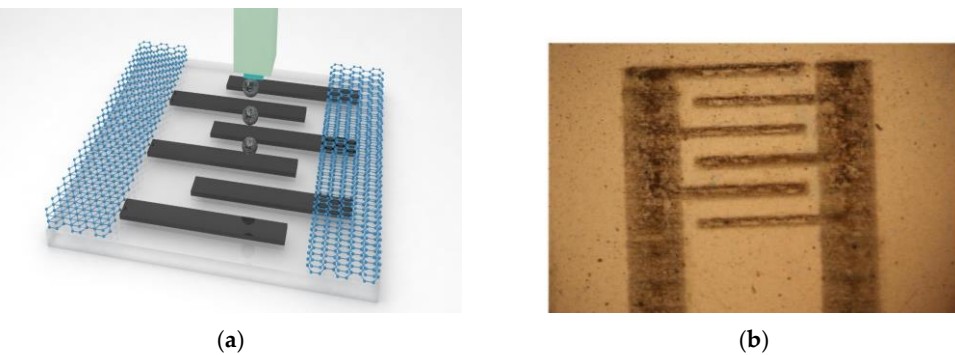

(**a**)                                                                                    (**b**)

**Figure 2.** (**a**) Printing graphene ink as a photoactive layer on top of the interdigitated electrodes. (**b**) Image of graphene electrode.

For the photodiode fabrication, an ITO glass slide was used as purchased and fixed on the printing platform. The ITO was used as a bottom electrode for the IT0/n-type $MoS_2$/graphene device, as shown in Figure 3. Then n-type $MoS_2$ ink was purchased from Graphene Laboratories Inc. (Graphene Supermarket, Ronkonkoma, NY, USA), and utilized without any further modifications. The ink was injected onto the ITO substrate using the same printer. The n-type $MoS_2$ ink was used as an active layer for the detection of the input light. Thermal annealing of inkjet-printed $MoS_2$ was performed at 300 °C to prevent oxidation of the printed material. Then graphene ink was deposited as a top electrode using the same printing steps.

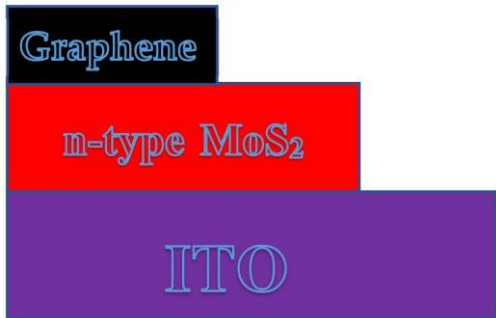

**Figure 3.** Structure of photoconductive diode.

*2.3. Characterization*

SEM (FEI Nova Nano SEM 230, Hillsboro, OR, USA) and TEM were used to examine the sample's surface and morphology to draw conclusions about the electrodes' structure and interface. The Titan transmission TEM (FEI Titan 80–300 kV, Hillsboro, OR, USA) operating at 300 keV was utilized to capture the TEM image. The X-ray diffraction spectra were acquired using a Philips/ PANalytical X-Pert PRO diffractometer (Fountain Valley, CA, USA), and the photoluminescent spectra were taken at a wavelength of 550 nm (PANalytical, Fountain Valley, CA, USA). A semiconductor characterization system (Keithley, Model SCS-4200, Cleveland, OH, USA) aided by a probe station was used to analyze the electrode electrical parameters.

## 3. Results and Discussion

*3.1. Graphene Ink Formalization*

The mechanical and electrical characteristics of the conductive patterns are greatly influenced by the ink solution's composition. Graphene inks that are the result of this process should be resistant to precipitation so that their performance remains consistent, and their conductive patterns remain homogenous. Fillers, surfactants, and additives should be selected for optimal processing compatibility while formulating graphene inks. For current printing technology to function optimally, printable ink needs to have certain fluidic qualities, such as optimal viscosity and surface tension. DMF was used to exfoliate graphene from graphite flakes, and then terpineol was distilled in place of DMF due to the considerable gap in their boiling points. Since terpineol has a much smaller volume than DMF, graphene can be concentrated quite a bit if the two solvents are mixed.

Polymer stabilization has been the primary focus of our work to improve the ink's formulation. As a precaution against the graphene flakes sticking together during distillation, a little amount of polymer (ethyl cellulose) was added to the recovered graphene/DMF dispersion. A quick anneal (baking on a hot plate at 300–400 °C in the air for roughly 1 h) after printing efficiently removed the stabilizing polymers. We have developed graphene/terpineol dispersions with a dilution of about 1 mg mL$^{-1}$ that are polymer-stabilized for a considerable duration. Ethanol was added to the dispersions just before printing to adjust the viscosity and surface tension so that they work with inkjet printers.

### 3.2. Graphene Ink Characterization

The surface tension of the developed ink was measured using a KRÜSS drop-shape analyzer (DSA-1000) (KRÜSS, Hamburg, Germany). Surface tensions around 25 and 50 mN/m are indicated for use in printing processes, as they facilitate strong material adhesion [24]. By using ethyl cellulose as stabilizing polymer, we were able to generate a new graphene ink based on ethanol, a solvent that is safe for the environment, through the process of solution-phase exfoliation of graphite. The produced graphene ink bonds well to glass substrates and is very compatible with inkjet printing thanks to its low surface tension (31 mN/m) at room temperature (Figure 4). The viscosity at room temperature was measured by RheoSense m-VROC viscometer (San Ramon, CA, USA), and was found 7.5 cP at different shear rates. Since the inkjet printer we utilized during the entire fabrication process can only jet out ink or solution with a viscosity between 2 and 12 cP, the measured viscosity is suitable for usage with the printer.

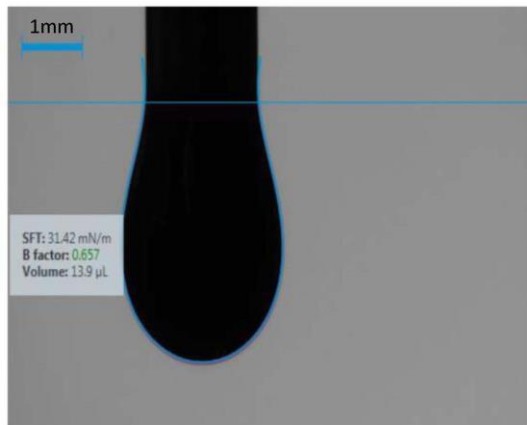

**Figure 4.** Graphene ink droplet.

### 3.3. Inkjet-Printed Electrodes

The fabricated interdigitated electrode is shown in Figure 2. Inkjet printing techniques to create thin film printing are crucial to determining the best substrate for flexible device development. The low optical reflectance of glass makes it a common material choice for optoelectronic devices. Ethanol and deionized water are used to remove any contaminants from the specified substrate. The clean glass is then placed on the printing platform of the inkjet printer. One of the most crucial steps in making stable and uniform MLG ink is finding the perfect balance for the ink's viscosity. These viscosities and concentrations have been tested and found to be suitable for use with an aerosol jet printer's ultrasonic atomizer. If the viscosity is too high (10 cP or higher), the printer will jam, and if it is too low (5 cP or lower), no droplet will form. The concentration and dispersing agents also have a significant impact on the ink's viscosity. Therefore, it is crucial to strictly regulate the ink production procedure since it has a major impact on the printability and quality of the printed patterns. In this work, the graphene ink viscosity was set to 7.2 cp and then injected on the substrate and annealed at 400 °C to fabricate multilayer graphene (MLG) electrodes.

Figure 5 shows the XRD pattern of a 20-layer graphene sheet. The pattern exhibits a characteristic peak at 2θ = 23.5°. The broadness of this peak may be due to the varying sizes of the graphene flakes and representing the disordered or highly amorphous nature of the graphene sheets. Moreover, this peak matches with the (002) peak of the graphene crystal structure that typically appears at 2θ = 26.5° and is available in the JCPDS Card No. 41-1487 [25]. Furthermore, the sharp peak at 2θ = 27.5° corresponds to the (100) peak and indicates the presence of larger graphene flakes or well-ordered regions of the film. For Cu Kα radiation XRD setup with a wavelength of 1.544 Å, the interplanar spacing d for the

peaks at 2θ = 23.5° and 2θ = 27.5° are calculated by Braggs Law and found to be 0.336 nm and 0.327 nm, respectively.

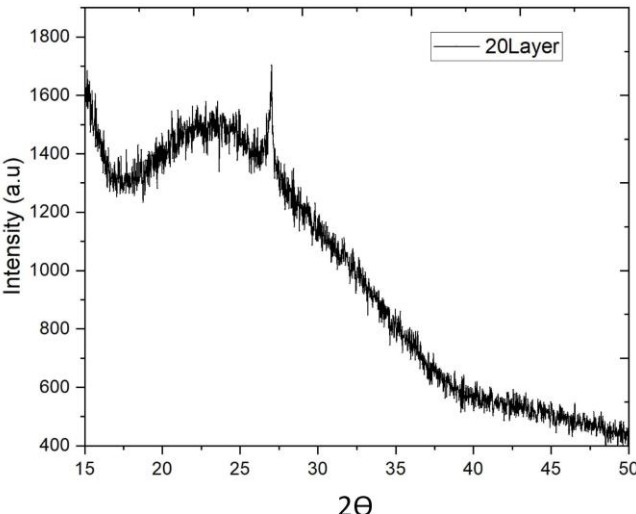

**Figure 5.** XRD spectrum of graphene powder.

The SEM image in Figure 6 shows uniformly distributed protruding graphene flakes with lateral dimensions. Low (Figure 6A) and intermediate (Figure 6B) magnification observations reveal the lateral contact between the graphene layers and the glass substrate. We can estimate the thickness of the graphene coating at ~500 nm. The SEM images display the edge of the glass substrate on top of which the graphene flakes have been deposited. Big particles are evident either on the graphene surface or on the lateral side. Figure 6C shows the SEM image of pristine $MoS_2$, and Figure 6D shows the SEM image of the $ITO/MoS_2/graphene$ device. The TEM analysis shown in Figure 7 reveals the presence of graphene flakes of different sizes. The structure also contains voids and particle agglomeration in some areas.

*3.4. Electrical Characterization of Electrodes*

The electrical properties were measured by the standard four-probe method and are shown in Figure 8. The MLG ink has low sheet resistance in comparison to dark ink (available in the market). Figure 8b reveals that the sheet resistance decreases with the increasing number of deposited MLG layers as shown by Equation (1), where $\rho$ is the resistivity of deposited ink, and t is the thickness of graphene layers. The maximum conductivity was achieved by printing 14-layer passes of ink. Beyond that, the sheet resistance increases and results in lower conductivities.

$$R = \frac{\rho L}{A} \ and \ A = wt \tag{1}$$

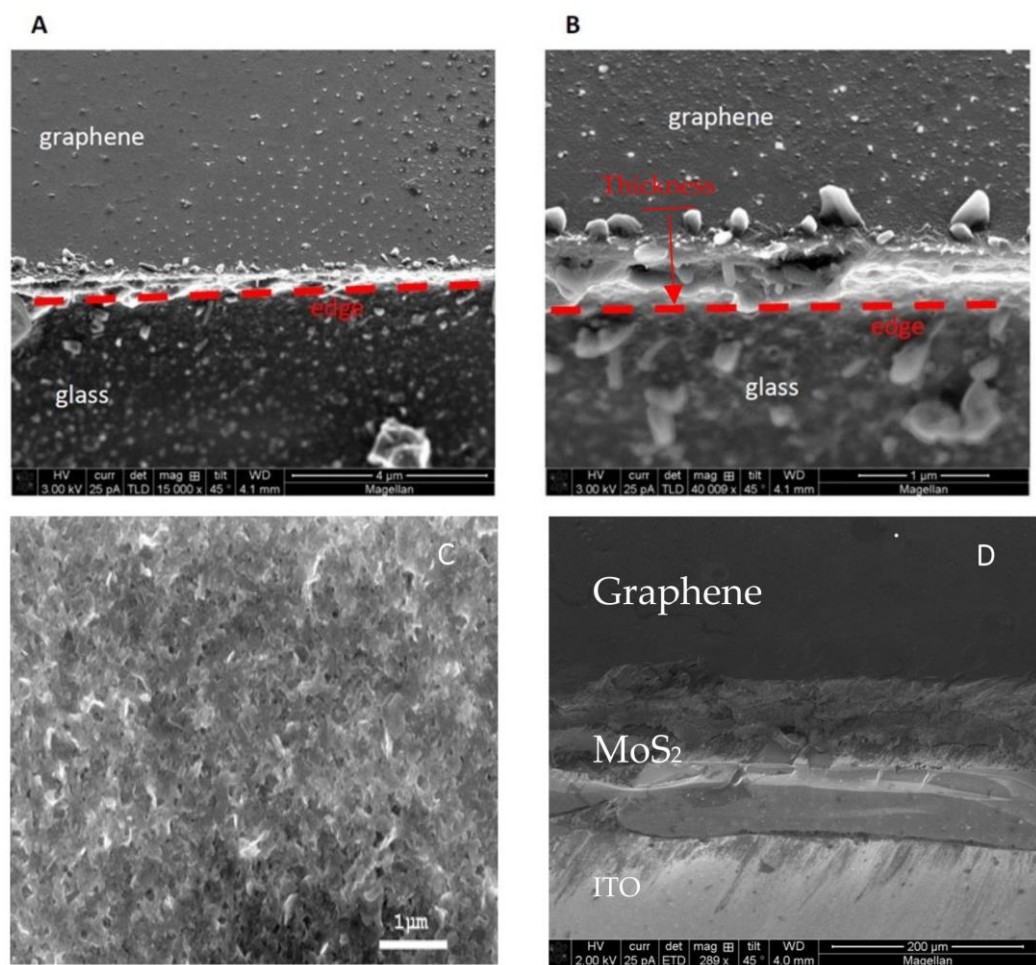

**Figure 6.** SEM characterization of graphene flakes (**A**,**B**); MoS$_2$ (**C**), reprinted from [26]; and the ITO/MoS$_2$/graphene device (**D**).

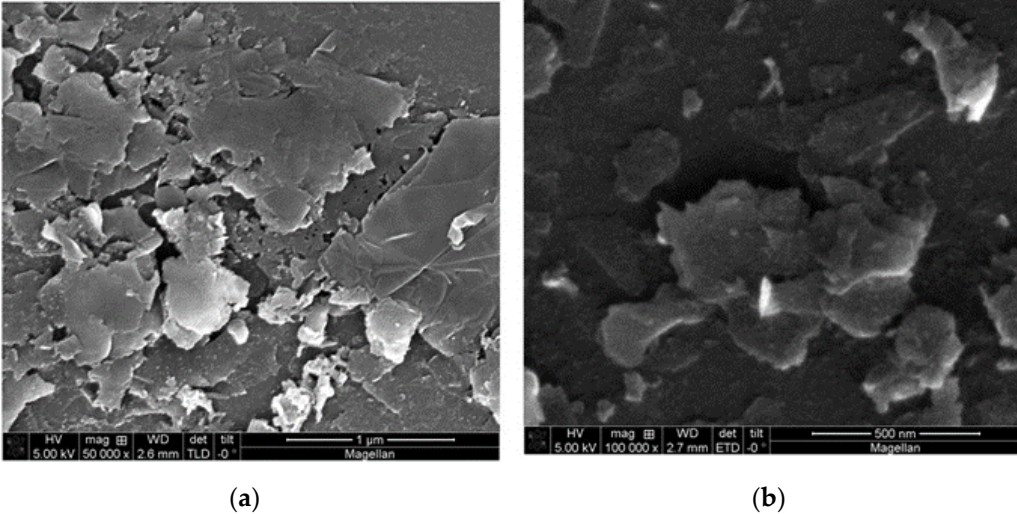

**Figure 7.** TEM images of graphene flakes of different sizes of different magnifications, (**a**) 1 μm resolution (**b**) 500 nm resolution.

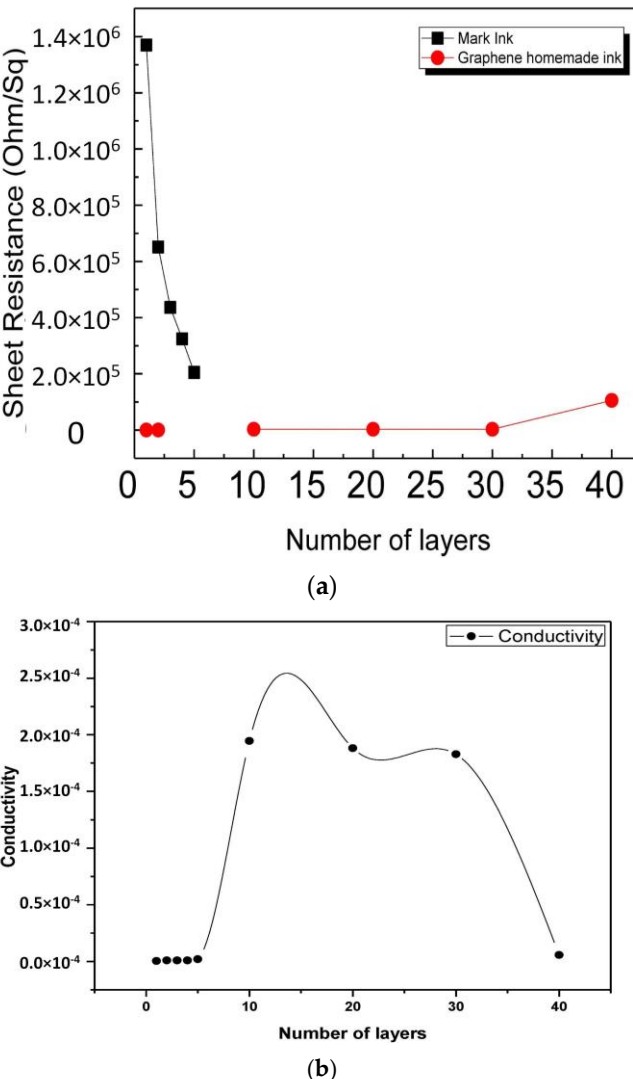

**Figure 8.** (**a**) Comparison of commercial graphene ink vs. homemade graphene ink. (**b**) Conductivity (siemens/cm) vs. number of layers.

It has been reported that graphene conductivity and flake quality both significantly degrade with prolonged sonication [27]. Although thin graphene flakes stack well, their high contact count reduces their electrical conductivity and may even increase their resistance. It is possible to reduce the number of interfaces between graphene flakes by using thicker flakes; however, this does not guarantee good stacking, and the printed material will have many voids. Flake thickness selection is a key factor in maximizing conductivity, which is important since ink applications degrade when connection breakdown is high [28].

Moreover, this phenomenon can be explained with the help of percolation theory [29]. According to this theory, the conductivity ($\sigma$) of the composite depends on the concentration of charges in random geometries, i.e., $\sigma \propto (\rho - \rho_c)^n$, where n is the percolation exponent. As seen in Figure 8b, the conductivity is very low for the initial five passes of graphene layers and does not form a conducting network. However, after achieving the percolation threshold ($\rho_c$) by increasing the layers of printed ink, conductivity is observed. In many cases, this conductivity behavior is attributed to network thickness "$t$", rather than the concentration of charges available at site, and is represented as $\sigma \propto (t - t_c)^n$, where $t_C$ is the critical thickness and depends upon the number of print passes [30]. The percolation threshold is the loading at which an electrically conducting network forms for a uniformly dispersed network. There is an abrupt increase in the composite's conductivity when the graphene particle loading exceeds $\rho_c$, i.e., five passes of printed graphene layers. It can

be also seen that the conductivity deceases after 25 passes of graphene layers. This is due to the disconnections and voids in the printed pattern after repeated annealing with temperature, which resulted in low conductivity.

The printed graphene pattern's conductivity is extremely important for optoelectronic applications. It has been studied that graphene electrodes performed well in the realization of photodiodes. Based on the above results, the MLG electrode efficiently harvests incident light and exhibits charge recombination and separation characteristics, thereby resulting in a cost-effective and improved electrode for photodiode applications. The device characteristics can be further improved to cover entire range of visible spectrum by carefully adding more graphene layers with a low-temperature inkjet printer technique.

### 3.5. Electrical Characterization of IT0/n-Type MoS₂/Graphene Device

The I–V characterization of IT0/n-type $MoS_2$/graphene device is evaluated at $500\,W/cm^2$ under bias voltages of +3 V and −3 V. As reported in the literature, carbon-based materials (graphene, in this case) usually act as p-type materials or create Schottky junctions with semiconductor materials [31,32]. Moreover, the work function of graphene is also tunable and reported to vary between 5.2 and 4.5 eV by the CVD approach, whereas the work function of ITO is between 4.0 and 4.5 eV [33]. The result depicted in Figure 9a also reveals a p–n junction formation at the intersection of the p-type graphene electrode and n-type $MoS_2$ active layers. In contrast, the ITO electrode formed an ohmic contact. The device showed the rectification characteristics of a standard p–n junction diode when +ve voltage was applied to the p-type contact. This indicated good Van der Waals interaction between *p*-type graphene and *n*-type $MoS_2$ heterojunctions with good current-rectifying behavior. When the energy of the excitation photon is greater than the bandgap of $MoS_2$ or the graphene layers, the electrons will be excited from the VB to the CB, resulting in the photocurrent. An illustration of band energy levels of ITO/$MoS_2$/graphene structure is shown in Figure 10.

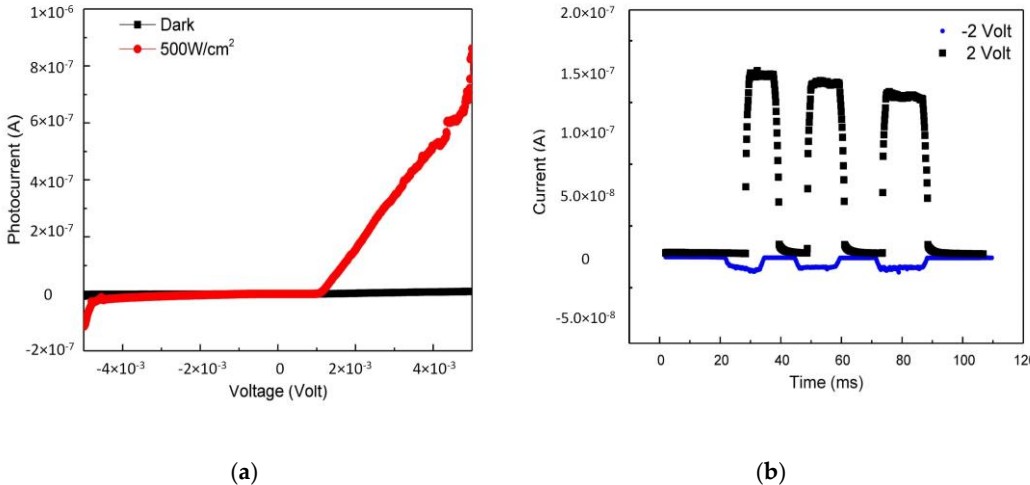

(**a**)                    (**b**)

**Figure 9.** (**a**) I-V characteristics of IT0/n-type $MoS_2$/graphene under $500\,mW/cm^2$. (**b**) ON-OFF characteristics of a device.

The ITO/$MoS_2$/graphene structure is a heterojunction between semiconductors (ITO and $MoS_2$) and a semimetal (graphene). $E_{g1}$, $E_{g2}$, and $E_f$ represent the band energy gap of ITO, graphene, and Fermi energy levels, respectively. ITO generally has a relatively high work function with a bandgap of around 3.6 eV, $MoS_2$ has a bandgap of 1.8 eV, and graphene is a zero-gap semiconductor with a Dirac point at the Fermi level. In the dark mode, the Fermi energy of graphene is located at $E_f$, while the Fermi energy of $MoS_2$ and ITO lies within the valence band. When light is incident on the heterostructure, it may excite trap electrons from the valence band to the conduction band of $MoS_2$, creating an electron flow. These electrons can then pass through the potential barrier at the $MoS_2$/graphene interface and enter the semimetal (graphene) layer. As a result, the Fermi energy of graphene shifts

upward toward the conduction band. The probability of electron–hole pair formation in graphene increases as the density of states (DOS) of the semimetal at the Fermi energy increases. This results in an increase in the photocurrent.

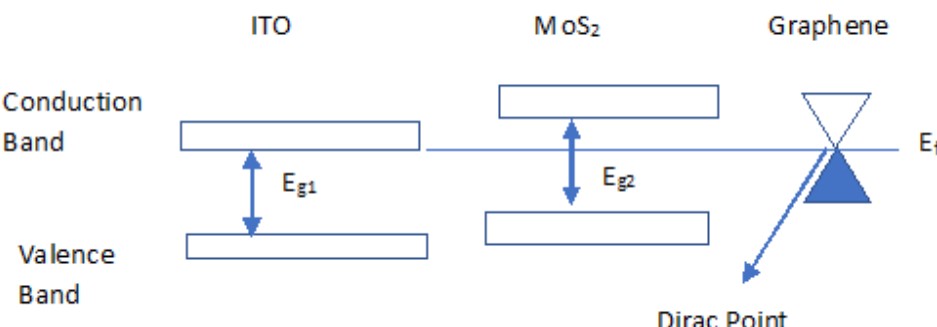

**Figure 10.** Illustration of band energy levels of the ITO/MoS$_2$/graphene structure.

　　　It should be noted that for negative bias, the device shows higher resistance with very low conductivity, thus confirming the behavior of the diode. Figure 9b depicts a high ON–OFF ratio for incident light. The photocurrent under illumination rises quickly to 15 μA for +2 V and becomes negligible or zero for dark conditions. Moreover, the current at −2 V under lighting is equal to the dark current, which depicts the reverse bias condition of the standard P–N junction diode. Furthermore, a decay in the magnitude of photocurrent has also been observed and attributed to traps. This is believed to be due to traps caused by the interaction of the MoS$_2$ flakes with the graphene electrode or the ITO substrate. The slow decay in the photocurrent is due to trap refill and thermal detrapping of charge carriers [34].

　　　Figure 11 depicts the photocurrent behavior under different wavelengths and applied bias voltages. There is no significant change in photocurrent response for negative bias voltages. A decaying photocurrent response is observed for positive bias voltage apart from two wavelengths, i.e., λ = 450 nm and λ = 575 nm. The dip or low photocurrent equal to dark current can be seen at λ = 450 nm. Since MoS$_2$ material is a strong light absorber at this wavelength, the device's response is attributed to this behavior.

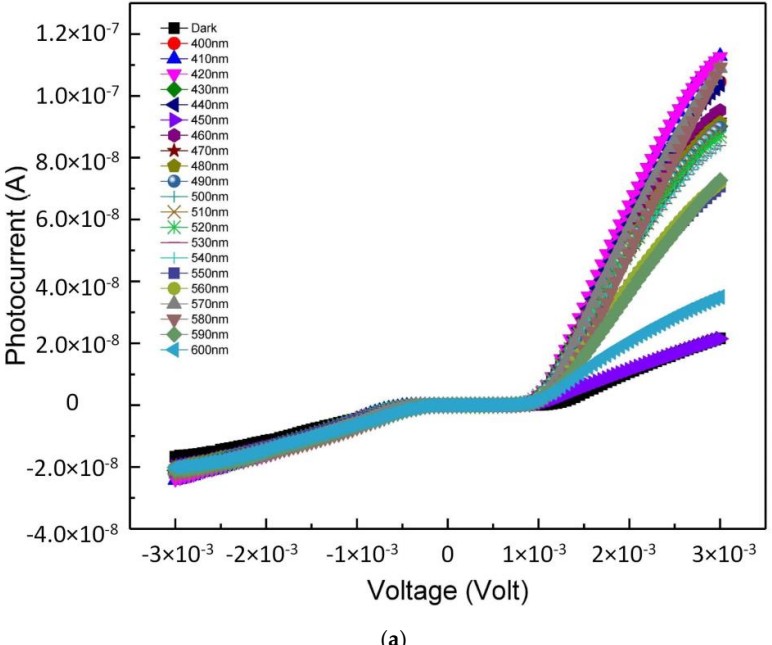

(**a**)

**Figure 11.** *Cont.*

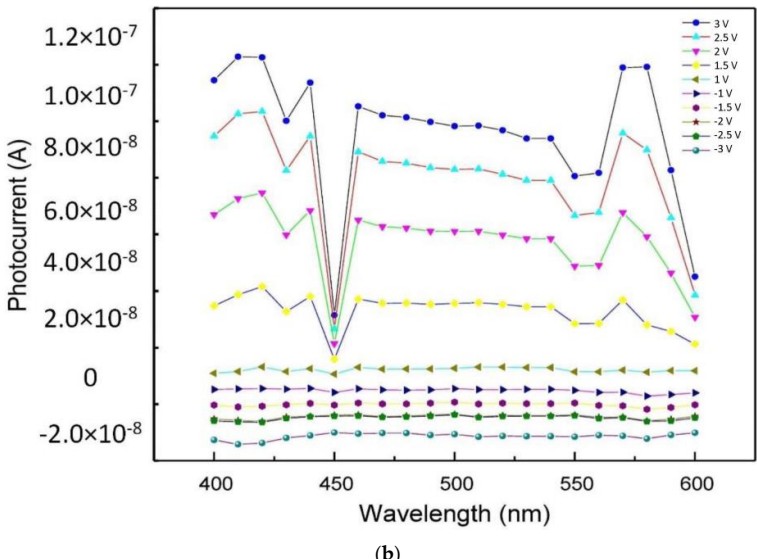

(**b**)

**Figure 11.** Photocurrent spectral response with respect to (**a**) different wavelengths and (**b**) applied bias voltages.

Moreover, the recombination of photoelectron–hole pairs is a severe problem associated with $MoS_2$ semiconducting material [35]. The fast recombination of charge carriers at this peak may result in exciton states and leave no free charge carriers for conduction. Moreover, the rise in photocurrent at $\lambda$ = 575 nm may be due to the release of charge carriers from excitons after some time and possible traps in the device.

## 4. Conclusions

In summary, we developed a multilayered graphene electrode using a low-cost inkjet printing technique. The sheet resistance and conductivity of electrode were measured for various print passes of layers. It was found that the conductivity initially increased and then decreased after 14 print passes of graphene layers. This effect was explained through the percolation theory arising from possible graphene ink agglomeration, cracks, and voids in the structure of the material. Then we fabricated an $ITO/MoS_2/graphene$ photodiode that showed good rectifying action under visible light region. The photodiode also showed a high ON/OFF ratio, and we expect that it can be used in high-speed optoelectronic applications. The inkjets exhibit outstanding jetting performance and consistently produce patterns at high resolutions of a few micrometers. Hence, the efficiency and quality of production will be greatly enhanced by our inkjet printing technology for 2D materials using a low-cost, scalable device. Therefore, it holds great potential for the development of organic and printed electronics.

**Author Contributions:** Conceptualization, A.M.A.-A.; methodology, A.M.A.-A., and N.E.I.B.; validation, T.K.N., and B.S.O.; formal analysis, A.M.A.-A.; investigation, A.M.A.-A., and N.E.I.B.; resources, B.S.O.; data curation, N.E.I.B.; writing—original draft preparation, A.M.A.-A.; writing—review and editing, T.K.N.; visualization, B.S.O.; supervision, A.M.A.-A.; project administration, A.M.A.-A.; funding acquisition, B.S.O. All authors have read and agreed to the published version of the manuscript.

**Funding:** This research work was funded by the Institutional Fund Project under grant no. (IFPHI 784-665-1442). Therefore, the authors gratefully acknowledge financial support from the Ministry of Education, King Abdulaziz University, DSR, Jeddah, Saudi Arabia.

**Institutional Review Board Statement:** Not applicable.

**Informed Consent Statement:** Not applicable.

**Data Availability Statement:** All the data is available in the article.

**Acknowledgments:** The authors gratefully acknowledge technical support from the Ministry of Education, King Abdulaziz University, DSR, Jeddah, Saudi Arabia, in addition to King Abdullah University of Science and Technology (KAUST), Thuwal, Saudi Arabia. The authors would like to thank Alessandro Genovese for his help in performing the SEM/TEM images.

**Conflicts of Interest:** The authors declare no conflict of interest.

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
