# Peer review of "Printed Graphene Electrode for ITO/MoS2/Graphene Photodiode Application"

_coatings, doi:10.3390/coatings13050831_

Round 1

Reviewer 1 Report

The presented manuscript focuses optimization of the conductivity of graphene electrodes. This is an important property for practical applications of ITO/MoS2/graphene devices. However, the novelty and added value of the presented work remains insufficiently justified as there were lots of attempts to use inkjet printing for graphene-based materials for a variety of applications, and thickness-dependent electrical conductivity were already discussed in detail.

 Inkjet printing for graphene-based materials should be better discussed in the Introduction part showing benefits and challenges and proving the added value of the presented manuscript. Some important works on the subject:

https://www.nature.com/articles/s41598-019-40547-0

https://doi.org/10.1080/14786435.2020.1766710

https://doi.org/10.1016/j.physleta.2015.06.063

Additionally:

1.      Why only 5 layers were deposited using Mark Ink (Figure 8 (a))?

2.      Figure 8b. What are the dimensions of conductivity?

3.      Page 8. The sentence at 275 line: „Figure 8b reveals that the sheet resistance decreases with increasing the number of deposited MLG layers...“. However, if to look at Figure 8a sheet resistance increases then the number of layers changes from 30 to 40 from 0.0E+00 to 1.0E+0.5(Ohm/Sq)

4.      General question: how Figure 8b was received as if to look to Figure 8a sheet resistance for 10 to 30 number of layers equal 0.0E+00 (Ohm/Sq)?

5.      The quality of Figure 10a is rather poor.  

Author Response

First of all, we would like to thank the honorable editor and reviewers for their excellent suggestions and comments. Without their efforts, our manuscript would not have been in its current form. We have carefully revised the manuscript by considering all valuable suggestions and comments of the respected reviewers. We admit that the suggestions given by the reviewers helped us to address the technical aspects of the manuscript. We hope that the current revision is up to the standards of all reviewers. The detail of our rebuttal is given below. 

Reply to reviewer 1 comments (Updated with red color fonts) 

Serial No 

Comment 

Reply  

1 

The presented manuscript focuses optimization of the conductivity of graphene electrodes. This is an important property for practical applications of ITO/MoS2/graphene devices. However, the novelty and added value of the presented work remains insufficiently justified as there were lots of attempts to use inkjet printing for graphene-based materials for a variety of applications, and thickness-dependent electrical conductivity were already discussed in detail.

Inkjet printing for graphene-based materials should be better discussed in the Introduction part showing benefits and challenges and proving the added value of the presented manuscript. Some important works on the subject:

https://www.nature.com/articles/s41598-019-40547-0

https://doi.org/10.1080/14786435.2020.1766710

https://doi.org/10.1016/j.physleta.2015.06.063

Before an eventual resubmission, I would suggest that the authors improve the quality of both presentation and the text. They should also emphasize more clearly what is the novel contribution to a crowded field, and what can be learned from the study.

Thank you for this comment; it improves the clarity and manuscript quality.

In the introduction section, benefits, challenges, and novel contributions of inkjet printing techniques are added and highlighted in red font.

In the mentioned literature, radio-frequency (RF) magnetron sputtering (base pressure: 3.0 × 10−4 Pa) with an Ar plasma and Langmuir–Blodgett (LB) techniques were utilized for the deposition of materials. Both techniques were complex and costly.

In previous literature, inkjet printing techniques were mostly used to deposit pristine or mixed graphene as an active layer for electronic applications. In some cases, graphene electrode was also deposited but utilized for different applications. In this case, we have solely used graphene as an electrode for photodiode application. We have also focused on making the ink with low surface tension and the desired viscosity by exfoliating graphene nanoparticles to achieve good precision and resolution. Our technique provides a low-cost and scalable solution for future photodiode applications.

2

Why only 5 layers were deposited using Mark Ink (Figure 8 (a))?

Thank you for this comment.

It can be seen that the decrease in sheet resistance was negligible by increasing the number of layers of mark ink. Hence, it was assumed that the resistance of the mark increase was significantly greater than that of the graphene ink, and further investigation was not carried out.

3

Figure 8b. What are the dimensions of conductivity?

Thank you for this comment.

The unit of conductivity used in Figure 8b is Siemens/cm and is highlighted in red font.

4

Page 8. The sentence at 275 line: „ Figure 8b reveals that the sheet resistance decreases with increasing the number of deposited MLG layers...“.

However, if to look at Figure 8a sheet resistance increases then the number of layers changes from 30 to 40 from 0.0E+00 to 1.0E+0.5(Ohm/Sq)

Thank you for this comment.

The author is right that sheet resistance increases for layers from 30 to 40. Figure 8b reveals both trends of decrease and increase in sheet resistance with the number of layers of graphene.  The relevant text is highlighted in red font.

5

General question: how Figure 8b was received as if to look to Figure 8a sheet resistance for 10 to 30 number of layers equal 0.0E+00 (Ohm/Sq)?

Thank you for this comment.

The sheet resistance value of graphene ink was much lower than that of mark ink. The value of mark ink is in the order of 106, whereas for graphene ink, it is in the order of 103. Hence, it is difficult to reveal the values of graphene ink in the graph for 10 to 30 layers.

6

The quality of Figure 10a is rather poor.

Thank you for this comment, as it improves manuscript figure quality and understanding.  

Figure 10a quality is improved now.

Reviewer 2 Report

Here, the authors deposited MoS2 ink on the same pattern and used it as an active layer. They put MoS2 ink on ITO glass substrate and then deposited graphene ink as a top electrode to fabricate ITO/MoS2/graphene device. The fabricated device showed good rectification behavior and high ON/OFF switching behavior with max photocurrent of 15 µA at +2V.

This is a comprehensive study and worth publication after some minor revisions.

1. The XRD pattern should be compared with the standard card, corresponding standard peaks should also be presented and compared.

2. Unit of conductivity should also be presented in Figure 8.

3. Printable graphene can also be applied for the preparation of graphene/QD heterostructure photodetectors (10.1002/adom.202201889), which is of great significance and should be addressed.

Author Response

Reply to reviewer comments (Updated with purple color fonts) 

Serial No 

Comment 

Reply  

1 

The XRD pattern should be compared with the standard card, corresponding standard peaks should also be presented and compared.

Thank you for this comment; it improves the manuscript quality. 

The XRD pattern is compared with the standard card, and corresponding standard peaks are also presented and compared. The changes are highlighted in purple font.

2.

Unit of conductivity should also be presented in Figure 8 .

Thank you for this comment, its improve the clarity of the Figure.

Unit of conductivity is now added in Figure 8.

3.

Printable graphene can also be applied for the preparation of graphene/QD heterostructure photodetectors (10.1002/adom.202201889), which is of great significance and should be addressed.

Thank you for this comment and appreciate the interest of reviewer.

We agree with the reviewer. The relevant data is included and cited. The changes are highlighted in purple font.

Reviewer 3 Report

This work proposes the fabrication of a Printed Graphene Electrode for ITO/MoS2/Graphene Photodiode Application. The general concept of this technique indeed opens a niche for low-cost electronics production; however, some critical issues need to be addressed and fixed. Here are some questions, comments, and recommendations.

1.       The authors should clearly define the dimension of the printed electrode, details of the ITO substrate (conductivity), the designing process, and the surface area of the printed electrode.

2.       It is suggested to show the reproducibility of ink production and its results in printing in different batches.

3.       The authors stated, “We have developed graphene/terpineol dispersions with a dilution of about 1 mg/mL that are polymer stabilized for a considerable duration. Including the optimization data in the supplementary material is suggested, and what is a considerable duration?

4.       The SEM and XRD mainly show graphene characteristics. How about the MoS2? It is suggested to show layer-by-layer deposition conditions in SEM for clarity. Please validate the morphological characteristics of the whole structure. It is advisable to use Raman spectroscopy measurement.

5.       The authors should also support their findings with some theoretical approaches. Such as by illustrating how the energy band level is changed against the photocurrent in this sequential structure: ITO/MOS2/Graphene.

Author Response

Response to the Reviewers 

First of all, we would like to thank the honorable editor and reviewers for their excellent suggestions and comments. Without their efforts, our manuscript would not have been in its current form. We have carefully revised the manuscript by considering all valuable suggestions and comments of the respected reviewers. We admit that the suggestions given by the reviewers helped us to address the technical aspects of the manuscript. We hope that the current revision is up to the standards of all reviewers. The detail of our rebuttal is given below. 

Reply to reviewer 2 comments (Updated with Green color fonts) 

Serial No 

Comment 

Reply  

1 

The authors should clearly define the dimension of the printed electrode, details of the ITO substrate (conductivity), the designing process,

and the surface area of the printed electrode.

Thank you for this comment; it improves the clarity and manuscript quality. 

The contact electrode was deposited with a square pad for the electrical contact with a different gap (150,200,250 µm) as the layout that designed interdigital electrodes using advanced design system (ADS) Software and highlighted as the green font. The surface area of the electrode was taken as the product of the length (L) and width (W) of the electrode, whereas L= 3.72 mm and W = 0.57 mm.

2 

It is suggested to show the reproducibility of ink production and its results in printing in different batches.

The authors thank the reviewer for this comment.

The authors apologize to the reviewer regarding this comment. Due to limited time, we cannot reproduce ink in different batches at the moment.      

3 

The authors stated, “We have developed graphene/terpineol dispersions with a dilution of about 1 mg/mL that are polymer stabilized for a

considerable duration. Including the optimization data in the supplementary material is suggested, and what is a considerable duration?

The authors thank the reviewer for this comment, as it makes the figure easy to understand.

In order to obtain the best results, we utilized a well-known ink formulation method. A mixture of graphite in DMF solvent is sonicated for a given period. The resultant suspension will be centrifuged to obtain thick flakes in the sediment, and the supernatant will be harvested and added to the ethyl cellulose.

Terpineol was then added to the graphene/DMF solution. DMF is exchanged for terpineol through a rotary evaporator with a vacuum distillation technique. After DMF boils off, the remaining graphene ink can be harvested separately. A small amount of ink solutions can be diluted with terpineol for optical absorbance studies to check the graphene concentration. For printing, the graphene ink will be mixed with ethanol to optimize the viscosity for inkjet printing. The duration for making the final ink is one hour which is highlighted in green.

The SEM and XRD mainly show graphene characteristics. How about the MoS2? It is suggested to show layer-by-layer deposition conditions in SEM for clarity. Please validate the morphological characteristics of the whole structure. It is advisable to use Raman spectroscopy measurement.

Thank you for this comm; it improves the clarity and manuscript quality. 

We mistakenly wrote that we had prepared the MoS2 ink. However, the ink was purchased from a Graphene supermarket in the USA. The corrected text is highlighted in green font. The characteristics of MoS2 materials can be found on the link (provided by the supplier).

https://www.graphene-supermarket.com/products/molybdenum-disulfide-mos2-pristine-flakes-in-solution-100-ml

Currently, we do not have a Raman spectroscopy measurement setup available at our Lab. Hence, we apologize for presenting any related data.

5 

The authors should also support their findings with some theoretical approaches. Such as by illustrating how the energy band level is changed against the photocurrent in this sequential structure: ITO/MoS2/Graphene.

Thank you for this comment, as it improves the quality of the manuscript.

The illustration and theory of energy band levels are now discussed in the manuscript and highlighted in green font.

Round 2

Reviewer 1 Report

The manuscript was improved following some of the reviewer's comments. However, a spelling check is still required. For example in some places authors use “BN, MnO2, and MoS2“ but in others „MoS2“.

Author Response

Reply to reviewer 1 comments (Updated with red color fonts) 

Comment 

Reply  

The manuscript was improved following some of the reviewer's comments. However, a spelling check is still required. For example, in some places authors use “BN, MnO2, and MoS2” but in others „MoS2“.

Thank you for this comment; it improves the manuscript quality. 

The spell check is done and corrected the errors. The changes are highlighted in blue font.

Reviewer 3 Report

The authors mostly omit all the recommendations. Just one suggestion regarding the energy band is fulfilled. The MoS2 physical characterization is lacking—no proof of successful layer-by-layer deposition is shown. Moreover, the reproducibility of the printing results must be proven. For this reason, the recommendation for this work is rejected. 

Author Response

Reply to reviewer 2 comments (Updated with Orange color fonts) 

Serial No 

Comment 

Reply  

1 

The authors mostly omit all the recommendations. Just one suggestion regarding the energy band is fulfilled.

i) The MoS2 physical characterization is lacking—

ii) no proof of successful layer-by-layer deposition is shown.

iii) Moreover, the reproducibility of the printing results must be proven.

Thank you for this comment; it improves clarity and manuscript quality. 

i) The physical characterization of MoS2 is now included in Figure 6.

ii) The conductivity graph in Figure 8 depicts the successful layer by layer deposition of graphene. The conductivity changes as the number of layers increases. The change in conductivity is also supported by percolation theory.

iii) The printing results were performed 3 times and reproducibility of results were achieved.